# The Effect of the Full Coverage of Essential Medicines Policy on Utilization and Accessibility of Primary Healthcare Service for Rural Seniors: A Time Series Study in Qidong, China

**DOI:** 10.3390/ijerph16224316

**Published:** 2019-11-06

**Authors:** Ying Wang, Yulei Zhu, Hang Shi, Xiaoluan Sun, Na Chen, Xin Li

**Affiliations:** 1Department of Health Policy, School of Health Policy and Management, Nanjing Medical University, Nanjing 211166, China; irene0505@njmu.edu.cn (Y.W.); zhuyulei0611@njmu.edu.cn (Y.Z.); shihang@njmu.edu.cn (H.S.); sunxiaoluan0207@njmu.edu.cn (X.S.); 2School of Health Economics and Management, Nanjing University of Chinese Medicine, Nanjing 211166, China; chenna@njucm.edu.cn; 3Department of Clinical Pharmacy, School of Pharmacy, Nanjing Medical University, Nanjing 211166, China; 4Center for Global Health, School of Public Health, Nanjing Medical University, Nanjing 211166, China

**Keywords:** essential medicines, rural seniors, full coverage, interrupted time series

## Abstract

*Background*: Since 2015, in order to handle the increasing prevalence of age-related diseases and escalating health expenditures arising from the aging population, the full coverage of essential medicines (FCEMs) policy for rural seniors has been implemented in primary healthcare institutions of Qidong County of Jiangsu, China. The purpose of this study is to examine the long-term effects of the introduction of FCEMs’ policy on the utilization and accessibility of primary healthcare service for elderly beneficiaries. *Methods*: The retrospective study was conducted in Qidong County in the Jiangsu province, China. A 47-month longitudinal dataset involving 91,444 health insurance claims records of inpatients aged 70 and older in primary healthcare institutions was analyzed. Changes in health service utilization (average length of stay), patient copayments (out-of-pocket expenses), New Rural Cooperative Medical System (NRCMS) reimbursement rate and daily hospitalization costs per patient were analyzed using interrupted time series analysis. Augment Dicky-Fuller unit root method was used to test the stationarity of the series alongside the Durbin Watson method to test autocorrelation. *Results*: Average length of stay increased at 0.372 bed-days per month before the implementation of FCEMs policy, whereas the increasing trend was slowed down at 0.003 bed-days per month after the implementation of FCEMs policy (*p* < 0.001). The average out-of-pocket expenses increased by 38.035 RMB monthly in pre-implementation of the policy period, but it decreased at the rate of 5.180 RMB per month after the implementation of the FCEMs policy (*p* = 0.006). The NRCMS reimbursement rate increased at 0.066% per month in pre-implementation of policy and the increasing trend was sharper at 0.349% in post-implementation of policy (*p* = 0.135). The daily hospitalization costs per patient decreased by 6.263 RMB (*p* = 0.030) per month, whereas it increased at the rate of 3.119 RMB (*p* = 0.002) per month afterwards. *Conclusions*: Based on interrupted time series analyses, we concluded that FCEMs policy was associated with positive changes of average LOS and average OOP expenses. The FCEMs policy has alleviated the financial burden of the rural seniors and slightly improved the efficiency of primary health service utilization. However, it had no positive effect on daily hospitalization costs. Therefore, in the general framework of FCEMs policy, the Chinese health policy-maker should take necessary supporting measures to curb climbing hospitalization expenditures and promote the rational drug use in primary healthcare institutions.

## 1. Introduction

In 2017, a census by China′s Statistics Bureau reported that the number of people aged 60 years and older has risen to 241 million, or approximately 17.3% of the total population of China [1]. It is predicted that the Chinese aging population will grow more than threefold over the next several decades. If this progress continues, the United Nations (UN) estimates that aging population will reach to 400 million and account for one third of the national population in 2030 [2]. With the growth of the geriatric population, China is facing the increasing aging-society challenge. Particularly, increases of numbers of seniors in the population coupled with the increased chronic disease burden associated with aging will provide the Chinese healthcare system with a huge challenge of delivering cost-effective primary healthcare service [3]. Due to an increase in chronic diseases, such as diabetes, hypertension, and chronic obstructive pulmonary disease, there is a growing demand for continuous-use medicines to treat such diseases. Meanwhile, the basic medical insurances have covered more than 95% of the population, while the low level of quality care is unable to address the concerns that elderly with prevalent diseases, especially for rural inpatients. It has also been demonstrated that the serious burden for patients is attributable to escalating medical costs for the treatment of a few chronic conditions in the primary care setting [4,5].

A study conducted in western rural China revealed that high out-of-pocket (OOP) spending and low healthcare reimbursement rate led to medical impoverishment, which was an important influencing factor to poverty [6]. On the other hand, drug expenditure accounted for 42% of the total expenditure for public hospitals on average in 2009 [7]. In primary healthcare institutions, about two thirds of outpatient expenditure and half of inpatient expenditure were sourced from drugs. Moreover, in the past decades, drug expenditure continued to increase at an average annual rate of 14% [8]. With the increased drug expenditure, the financial burden of chronic diseases for the elderly grew rapidly. In order to slow down health expenditure growth, China carried out a new round of healthcare system reforms since 2009. National Essential Medicine Policy (NEMP), as one of the key components in new healthcare reform, was established to improve rational use of drugs and promote accessibility and affordability of medicines, and reduce the financial burden of patients. The concept of essential medicines proposed by the World Health Organization (WHO) referred to those who can satisfy priority health care needs of the population [9]. Although the implementation of the NEMP in primary health care facilities has been progressing, the availability of essential medicines is generally low and some essential medicines are unaffordable in China [10,11,12]. Less equitable access to essential medicines has been a formidable public health challenge in China, particularly for rural areas [13].

The full coverage of essential medicines (FCEMs) policy has been put forward to improve the availability and affordability of essential medicines worldwide. In accordance with a report of the Pharmaceutical Country Profiles (PCP), out of 105 countries of the WHO, 54 countries implemented the policy to promote free access to essential medicines [14,15]. In different countries, the FCEMs policy can be adopted in different forms such as essential medicines at no charge or copayment exemption for essential medicines. Recently, the effects of free access to medicines among the elderly people has been the subject of some empirical studies in Western countries. For instance, Puig-Junoy et al. evaluated the effects of coinsurance exemption for prescription drugs applied to the elderly after retirement in Spain and found that the zero copayment increased the prescription medicines expenditure on average by 17.5%, pharmaceutical spending by 25% and the costs paid by the insurer by 60.4%. However, the work did not find a significant decline in hospitalization rates for elderly people exempted from copayment [16]. Paniz VMV et al. examined the effects of free access to medicines for 4003 elderly inhabitants living in 41 Brazilian cities and proved that the free continuous-use medicines programs played a positive role in increasing availability of medicines among the elderly with chronic diseases [17]. In China, a series of FCEMs policies for psychosis, chronic diseases, infectious diseases and child vaccines were carried out since ten years ago. However, only a few studies investigated the effects of FCEMs policy. Most of these studies focused on the outcome indicators related with medicines at no charge directly, such as medication adherence or drug expenses [18,19,20]. These observational studies suggest that the full coverage strategy increases medication adherence of patients with chronic conditions [18,19,20], but its effects on actual healthcare spending and utilization have not been rigorously assessed. Chinese literature paid little attention to the effects of FCEMs policy on utilization and accessibility of the healthcare service for rural seniors. Usually, most research on the effects of FCEMs policy only adopted the pre–post design and incorporated short-term data in certain hospitals. Furthermore, few studies seemed to have commented whether the FCEMs policy benefits the rural elderly population by analyzing aggregate medical insurance data. More research works to assess the long-term effects of FCEMs policy is needed.

Qidong is a rural Chinese county, which is located in eastern Jiangsu province. Since 2015, the government of Qidong implemented the FCEMs policy for rural seniors aged 70 and older. Therefore, as an empirical study, the objectives of present study were: (1) to detect the long-term effect of FCEMs policy on primary healthcare service utilization and (2) to assess the long-term effect of FCEMs policy on hospitalization costs and economic burden of diseases in rural elderly. As FCEMs policy was implemented in primary healthcare institutions of Qidong County, rural seniors who enrolled in New Rural Cooperative Medical System (NRCMS) were the target population.

## 2. Materials

### 2.1. Settings

The study was conducted in Qidong County in Jiangsu Province, China. Qidong is located to the north of Shanghai, at the Yangtze River Estuary, on the east coast of China. It has a coastal area of 1208 km^2^, and had a population of 1.12 million at the end of 2016. With the elderly population accounting for 23.01% of the total population since 2014, Qidong County has faced with the challenge of aging population. In Qidong County, there were 43 primary health institutions in total. Approximately, 4.6 medical technicians and 4.17 hospital beds for every one thousand inhabitants are available in this county. Healthcare service was primarily provided by 2207 licensed physicians and 1938 registered nurses in 2017 [21].

### 2.2. FCEMs Policy for Rural Seniors in Qidong County

From the start of NEMP in Qidong County in 2009, all the essential medicines were included in the reimbursement directory of NRCMS. In order to further eliminate out-of-pocket (OOP) costs for elderly patients, the FCEMs policy was implemented since January 2015. The essential medicines covered free of charge were selected from National Essential Medicine List (NEML) based on the following standard, which is as follows: (1) the retail price of oral dosage forms was less than 20 RMB and (2) the retail prices of injection dosage forms was less than 10 RMB. According to the FCEMs policy documents that were issued by the government of Qidong County, only the rural seniors aged 70 and older were the target beneficiaries of the policy. As a result, the subjects of this study were limited to those elderly patients over 70 years old. For the elderly hospitalized patients aged over 70 years at primary health institutions, NRCMS provided full reimbursement proportion for these essential medicines. According to statistics of the local health authority, the consumption of these medicines accounted for more than 80% of the total consumption of essential medicines in primary health institutions between 2014 and 2017.

### 2.3. Study Outcomes

We conducted an interrupted time series study, and the impacts of FCEMS policy on utilization and accessibility of primary healthcare service were evaluated for the elderly beneficiaries. Five outcome indicators were analyzed: number of hospitalizations, average length of stay (LOS), daily hospitalization costs per patient, average OOP expenses and NRCMS reimbursement rate (Table 1). Generally, the number of hospitalizations and LOS are measured as indicators of the efficient use of health service. Significantly reduced LOS and increased admissions could be associated with greater healthcare service utilization [22]. Daily hospitalization costs per patient, OOP expenses and NRCMS reimbursement rate were selected as indicators for accessibility of healthcare service [23]. For these indicators, the related intervention was the implementation of FCEMs policy, which tried to reduce the health expenditure and out-of-pocket expenses and increase medical insurance reimbursements.

These five outcome variables are aggregated monthly data at the patient level. The number of hospitalizations was measured by the hospital admissions in a given month. LOS was measured by the average hospital bed-days of the elderly patients hospitalized in a given month. The hospitalization costs were calculated where total expenses for beds, drugs and medical examinations incurred throughout the hospitalization period. Daily hospitalization costs per patient were calculated by using the overall inpatient expenses divided by number of hospitalizations and then divided by the average LOS during a given month. OOP expenses referred to patient payments for deductibles, as well as cash outlays for the items not covered by NRCMS. Average OOP expenses were measured by using total hospitalization costs minus expenses covered by NRCMS per patient in a given month. Similarly, average NRCMS reimbursement rate was computed by using expenses covered by NRCMS divided by total hospitalization costs per patient in a given month.

### 2.4. Data Collection

In Qidong County, from 2014 to 2017, there were 43 primary health institutions, which included 7 rural community health service centers and 36 township hospitals. All of the 43 primary healthcare institutions in Qidong County were included in this study. Meanwhile, all of the raw NRCMS’ claims data on the inpatients aged 70 years or older in the 43 primary health institutions was retrieved from the database of the department of the new rural medical scheme in Qidong County. Monthly data from January 2014 to November 2017 were collected. The data analysis focused on the elderly patients’ demographic characteristics (sex and age), hospitalization information (diagnosis, admission date, discharge date, number of hospitalizations and hospitalization costs) and reimbursement status (OOP expenses and reimbursement of NRCMS). Totally, data related to 48,071 patients and 91,444 patient admissions were obtained, which included the following variables: sex, age, health institutions, NRCMS ID, diagnosis, date of admission to hospital, date of discharge from hospital, total hospitalization costs, total reimbursement of NRCMS and total OOP expenses. The average age of the patients was 79.31 (Standard Deviation = 6.18). Less than two thirds of patients (59.9%) were female. The data for average LOS, daily hospitalization costs per patient, average OOP expenses and NRCMS reimbursement rate were based on these patients.

Approval for this study was obtained from the Institutional Review Board of Nanjing Medical University.

### 2.5. Statistical Analysis

An interrupted time series (ITS) regression was employed to analyze the 47-month data. When it is difficult or impossible to find a control group, the ITS model is regarded as the quasi-experimental design to analyze the longitudinal effects of interventions [24,25,26,27]. By using ITS in statistical terms, we can assess whether an intervention led to a transient or long-term outcome [28]. In this ITS model, in order to provide independent tests, the variation within the data was segmented into three parts: (1) the slope in scores for the pre-FCEMs policy period, (2) the change in level for the post-FCEMs policy period, accounting for the pre-FCEMs trend, and (3) the change in slope from pre-FCEMs policy to post-FCEMs policy. Given that the implementation of FCEMs policy in January 2015, hence, we set this month as the beginning of policy intervention. The pre-FCEMs policy segment of ITS analysis was fit with an intercept and a variable estimating trend [29], then trends of outcomes changing in slope were monitored in post-FCEMs policy period and changes in level were observed from pre-FCEMs policy period to post-FCEMs policy period. The ITS model was formulated as following:(1)$$Y_{t}=\beta_{0}+\beta_{1}T+\beta_{2}D+\beta_{3}P+\varepsilon$$

In detail, Y is the main outcome indicator. T is a count variable indicating the time in months throughout the survey, assigning 1 to 47 in this study; D is a dummy variable representing the period before and after the FCEMS policy intervention, in which D = 0 for pre-FCEMS policy period and D = 1 for post-FCEMS policy period; P is a continuous variable counting the number of months after the FCEMs policy intervention, for the time before the FCEMs policy intervention P = 0. Furthermore, $\beta_{0}$ indicates the baseline level of the outcomes when T = 0; $\beta_{1}$ estimates the change of trends prior to the FCEMS policy; $\beta_{2}$ is the instantaneous level change in the implementation of the policy and $\beta_{3}$ is interpreted as the change in slope of trend in the outcome before and after FCEMS policy. Hereby, $\beta_{1}+\beta_{3}$ is denoted as the actual trend of the outcomes after the implementation, indicating the net effect of policy intervention. ε is the error term. The ITS model was shown in Figure 1.

We employed Augment Dickey-fuller (ADF) unit root test to identify the stationarity of series data [29]. The null hypothesis of ADF test is that there is a unit root at some level of confidence, which suggests non-stationarity of the time series data. The results of ADF statistics are negative numbers. The more negative, the more likely a rejection of the null hypothesis. In this study, if the statistics of the five indicators were less than 5% critical value, we could conclude that these time series data were stationary. Meanwhile, the Durbin Watson test was employed to test the first-order autocorrelation of the data [30]. It is accredited that when the Durbin Watson statistics are approaching 2, it is extremely possible the observations are independent. The feasible generalized least square (FLGS) method would be employed to modify the first-order autocorrelation errors [31,32,33]. Analyses were performed with STATA v.14 software (STATA Corporation, College Station, TX, USA) and a statistical significance level of 0.05 was set for rejecting the null hypothesis.

## 3. Results

The annual five indicators from January 2014 to November 2017 were shown in Table 2. The results showed an increasing trend on the daily hospitalization costs per patient during the study period. Meanwhile, the results showed a decreasing trend on average LOS and average OOP expenses, while there existed an opposite trend on NRCMS reimbursement rate.

Table 3 showed the results of the ADF unit root test. The statistics of five indicators were less than 5% critical value and suggested these series data were stationary (*p* < 0.05). As shown in Table 4, all of the Durbin-Watson statistics are approaching 2. Due to the corrections by FLGS method, there were no autocorrelations in the observations.

### 3.1. Primary Health Service Utilization

As Table 4 shows, a declining trend was found in the number of hospitalizations from month to month before the FCEMs policy, but this decrease was not significant (*p* = 0.227). After the FCEMs policy started, no abrupt change in level was noted, but change from decreasing to increasing in the trend can be observed (*p* = 0.034). The change in slope also showed an increase but not statistically (*p* = 0.238).

As shown in Table 4, the results of parameter estimation of ITS analysis on average LOS after the correction for autocorrelation and Figure 2b showed the time series of average LOS of the elderly in primary health institutions. The average LOS showed an increasing trend and the rate of the increase was 0.372 (*p* < 0.001) bed days per month prior to the implementation of the FCEMs policy, then the average LOS decreased by 3.257 bed days immediately (*p* < 0.001) at the beginning month of intervention. In post-FCEMs policy period, the upward trend of average LOS was slowed down and the rate of increase was 0.003 bed days per month (*p* < 0.001).

### 3.2. Daily Hospitalization Costs of the Elderly Inpatients

The results of the parameters estimation of daily hospitalization costs per patient were shown in Table 4 and time series of hospitalization costs per patient was illustrated in Figure 2b. Daily hospitalization costs per patient decreased at the rate of 6.263 RMB (*p* = 0.030) per month before the implementation of FCEMs policy. At the beginning month of policy intervention, an increase in level was observed and daily hospitalization costs per patient increased by 13.844 RMB (*p* = 0.523) immediately. However, the daily hospitalization costs per patient increased at the rate of 3.119 RMB (*p* = 0.002) afterwards.

### 3.3. OOP Expenses and NRCMS Reimbursement Rate

Table 4 showed the results of the financial burden of the elderly estimated by ITS model during the study period and Figure 2 illustrated the overall trend of NRCMS reimbursement rate and average OOP expenses. The NRCMS reimbursement rate increased at 0.066% per month in pre-FCEMs policy period and the increasing trend was sharper at 0.349% in post-FCEMs policy (*p* = 0.135). Average OOP expenses showed the opposite trend before and after the policy intervention. At first it increased steadily at the rate of 38.035 RMB per month (*p* = 0.012) prior to the intervention, and then an immediate drop in level was observed and the average OOP expenses descended by 626.905 RMB (*p* < 0.001) at the beginning month of policy intervention. However, the average OOP expenses decreased by 5.18 RMB per month (*p* = 0.006) after the implementation of FCEMs policy.

## 4. Discussion

Using 47-month series medical insurance data from January 2014 to November 2017, an interrupted time series analysis in this study evaluated the impacts of FCEMs policy on number of hospitalizations, average LOS, daily hospitalization costs per patient and patient copayments. The information provided in this work demonstrated the policy resulted in a decrease of the OOP expenses and an increase in daily hospitalization costs per patient, while contributed to a deceleration of the growth of LOS. Our results also suggested that the implementation of FCEMs policy in Qidong County had a slight effect on increasing number of hospitalizations and NRCMS reimbursement rate.

In this study, the number of hospitalizations and average LOS were employed to assess the efficiency of health care utilization, evidently, a shorter LOS has been associated with an increase of hospital expenses [34]. Due to free access to essential medicines in elderly hospitalized patients, we assumed that hospitalizations and LOS increased rapidly after FCEMs policy. As is expected, after the month of FCEMs policy, a significant transient rise in number of hospitalizations was observed. However, the policy generated little or no impact on increasing the number of hospitalizations. Meanwhile, two different findings were observed by the descriptive analysis and the ITS model, respectively. The findings of the ITS model showed that the FCEMs policy resulted in a deceleration of the growth of LOS in Qidong County. However, surprisingly, contrary to our hypothesis, the average LOS per month was reduced after the implementation of FCEMs policy based on the results of descriptive analysis. As the elderly outpatients were excluded from the beneficiary population, the beneficiaries of FCEMs policy were only the elderly inpatients. Compared with usual access to medicines, the elderly inpatients could take the essential medicines at no charge. Two potential reasons could be proposed to explain this finding. First, The FCEMs policy could enhance aged patients′ compliance with drugs and promote the efficiency of essential medicines in the treatment of chronic diseases. Some studies in Western countries claimed the free access to medicines contributed to a positive impact on medication adherence. For example, a study from the United States confirmed that the medication adherence of patients with myocardial infraction had been improved as a result of full coverage for preventive medications [35]. In general, improvements in medication adherence usually led to the achievement of longer-term therapeutic and outcome goals [34,35,36,37], especially for reductions in LOS [36,37]. Therefore, the deceleration of the growth of LOS could be associated with the implementation of FCEMS policy. Second, the LOS of patients with mild-to-moderate illness severity could be shorter than the patients with severe illness. Some previous empirical studies in China could demonstrate the relationship between the patients′ disease severity and LOS. For example, Li et al. demonstrated that the disease severity was the most important influencing factor for length of stay, and the more severe the disease of patients, the longer the length of stay [38]. Mo et al. also found that the disease severity had an association with length of stay [39]. According to our data analysis in this study, no significant change was observed in the elderly patients′ disease category before and after the FCEMs policy. Under the premise of reduction of LOS after the FCEMs policy, we can conclude that there were changes in the disease severity of elderly inpatients during the pre- and post-FCEMs policy period. After the implementation of FCEMs policy, in order to get essential medicines at no charge, more and more patients with mild-to-moderate illness severity were transferred to the inpatient units from outpatient clinics in primary health care institutions. Compared with the patients with severe illness, the LOS of the patients with mild-to-moderate illness severity could be shorter. Therefore, the changes in the disease severity of elderly inpatients led to reduction of LOS after the FCEMs policy.

The average OOP expenses increased before the implementation of the FCEMs policy, but it decreased afterwards. The finding of the impact of FCEMs policy on OOP expenses was similar with previous studies that claimed a reduction in OOP expenses owing to dispensing drugs free of charge [35,40,41]. It is considered that improved reimbursement rate driven by the FCEMs policy has the potential to save medical costs and reduce the financial burden of the elderly. For example, a study investigating patients with early breast cancer found that the full coverage of aromatase inhibitors had yielded greater health benefits at a lower cost [34]. In addition, another study concentrating on post-myocardial infraction patients suggested the full prescription drug coverage was able to save money over the long-term [42]. Furthermore, from a society perspective, this finding is encouraging as it is likely to improve health care equities and decrease the incidence of medical impoverishment. For instance, Xu et al. suggested that the provision of drugs at free charge to patients with hypertension resulted in reduced medical costs and financial burden, along with improved accessibility and equity of drugs [43]. While a study focused on chronic disease treatment indicated free access to essential medicines has been effective to the poor population and decreased inequalities, but with differences in regions and classes of medicines [44].

Another key finding of our study is that the NRCMS reimbursement rate had been increasing from 53.29% to 70.35% throughout the study period in Qidong County. NRCMS reimbursement rate was used to evaluate the accessibility of primary health service in this study. Usually, a higher health insurance reimbursement rate contributed to an improvement of accessibility of health service [45,46]. On one hand, the increasing NRCMS reimbursement rate indicated that FCEMs policy improved the accessibility of health service for rural seniors. Evidence demonstrated patients with lower reimbursement rate were less likely to receive healthcare service [47]. Due to the increasing NRCMS reimbursement rate, the expenses of health service were more affordable for the patients, who could stimulate the willingness of patients to seek more health service [48], particularly for the rural senior beneficiaries with low-income and low-health [49]. On the other hand, the increasing NRCMS reimbursement rate had the potential to lead to an unreasonable increase in reimbursement expenditure. Simultaneously, the reimbursement for other health-insurance beneficiaries could be restricted due to limited health care funding.

However, the FCEMs policy also resulted in a rapid growth of daily hospitalization costs per patient in Qidong County. Although the daily hospitalization costs per patient decreased before the implementation of FCEMs policy, it increased after the implementation of FCEMs policy. The positive relationship between health expenditure and reimbursement rate is well-established in previous literature [50]. We have hypothesized that the hospitalization costs could increase dramatically after FCEMs policy. Coincidentally, this finding suggested that the daily hospitalization costs per patient increased after the implementation of FCEMs policy. A complex set of policy, healthcare, technology, and economic factors can result in escalation of the daily hospitalization costs and it is unlikely to be addressed by FCEMs policy alone. One possible reason is that the FCEMs policy unleashed the rural seniors′ pent-up health demand in primary health care institutions. In general, the FCEMs policy had no positive effect on hospitalization costs. Empirical evidence indicated the irrational use of medical service will abound if there is a full coverage of health care. A national survey data indicated that health insurance coverage led to a significant effect on the possibility that the elderly would overuse health service [51]. From the perspective of service providers, to compensate for the loss due to decreased drug costs, health institutions have the possibility to increase expenses of treatment and examination, while physicians may induce patients, especially those with rural medical insurance, to receive excessive medical treatment [52]. Therefore, constant efforts are required to reduce unreasonable health expenses in Qidong County. First, the criterion for hospitalization should be established to control the rapid increase in unreasonable admissions, along with strengthened supervision of health care-seeking behaviors, and the physicians should be responsible for accurate suggestions on seeking service to avoid elderly patients taking up medical resources excessively [53]. Second, considering repetitive distribution over a short period to the elderly may cause waste of essential medicines, the frequency of distribution and classes of essential medicines need to be regulated effectively.

Meanwhile, as the strongest, quasi-experimental design [54,55], interrupted time series analysis was employed to evaluate the long-term effects of the FCEMs policy. Nevertheless, our study is subject to several other limitations. First, it lacks a control group to explore whether there are other factors, rather than the policy implementation, that result in the change of primary outcomes. Although it is not required for ITS analysis to establish associations between interventions and outcomes, a control group can help us better understand the impact of the interventions. Second, as a result of constraint of data, the outcomes in the study are not so comprehensive that we cannot assess the impact of the policy on medication adherence, clinical outcomes, irrational use of essential medicines and health equities. Third, limited to the setting in a county, the study should have been carried out in more representative municipalities, provinces and countries ideally. However, the FCEMs policy in infancy has no national criteria for selection and distribution of essential medicines and the beneficiaries depend on the decision of the local governments. Fourth, due to the local policy provisions, there is a restriction on age of beneficiaries in this study. Only the elderly inpatients aged 70 years or older are selected as the target population, limiting generalizability to the entire healthcare systems.

## 5. Conclusions

In summary, this study analyzed the changes in the number of hospitalizations, average LOS, daily hospitalization costs per patient, average OOP expenses and NRCMS reimbursement rate before and after the implementation of FCEMS policy in Qidong County of Jiangsu Province, Eastern China during the 47-month period from 2014 to 2017. Based on the interrupted time series analysis, we concluded that FCEMs policy was associated with positive changes of LOS. The FCEMs policy decelerated the growth of LOS and reduced average LOS per month simultaneously. Meanwhile, the FCEMs policy implemented at Qidong County seemed to have positive impacts on average OOP expenses and NRCMs reimbursement rate. It has alleviated the financial burden of the rural seniors and slightly improved the efficiency of primary health service utilization. However, the FCEMs policy had no positive effect on daily hospitalization costs. As a consequence, effective initiatives should be taken such as strengthen the supervision, improve rigorous selection and distribution of essential medicines at free charge to constrain climbing health spending.

## Figures and Tables

**Figure 1 ijerph-16-04316-f001:**
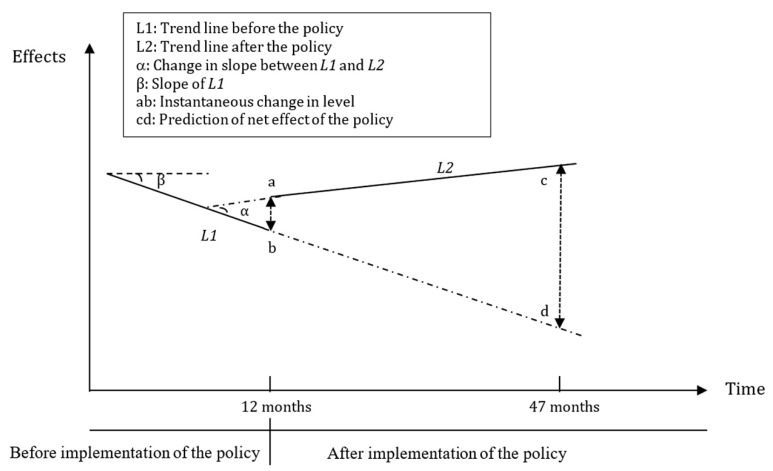
Graphic illustration of interrupted time series model and measuring the policy effects based on trend lines for data points before and after the policy.

**Figure 2 ijerph-16-04316-f002:**
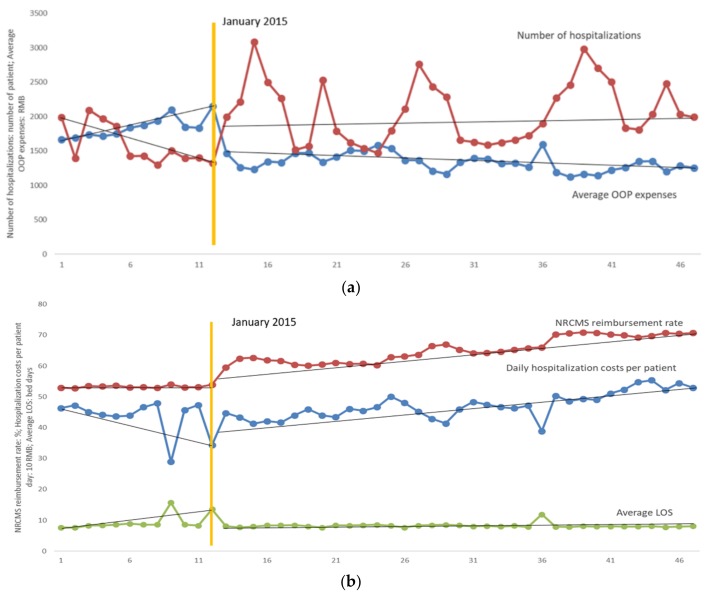
(**a**) Impacts of full coverage of essential (FCEMs) policy on average out-of-pocket (OOP) expenses and number of hospitalizations. (**b**) Impacts of FCEMs policy on average length of stay (LOS), New Rural Cooperative Medical System (NRCMS) reimbursement rate and daily hospitalization costs per patient day.

**Table 1 ijerph-16-04316-t001:** Study outcome variables.

Outcomes	Indicators	Units of Measurement
Utilization of primary healthcare services	Number of hospitalizations	Number of patients
	Average LOS ^a^	Hospital inpatient bed-days
Accessibility of Primary Healthcare Services	Daily hospitalization costs per patient ^b^	RMB
	Average OOP expenses ^c^	RMB
	NRCMS reimbursement rate ^d^	Percentages

**^a^** Average length of stay $=\frac{length\;of\;stay}{number\;of\;hospitalizaitons};$
**^b^** Daily hospitalization costs per patient $\;=\frac{total\;hospitalization\;costs}{number\;of\;hospitalizations*average\;length\;of\;stay}$; **^c^** Average out-of-pocket expenses $=\frac{total\;out-of-pocket\;expenses}{number\;of\;hospitalizaitons}$; **^d^** NRCMS reimbursement rate $=\frac{total\;reimbursement\;of\;NRCMS}{total\;hospitalizaiton\;costs};$ LOS, length of stay. OOP, out-of-pocket. NRCMS, new rural cooperative medical system. RMB, renminbi, Chinese currency.

**Table 2 ijerph-16-04316-t002:** Outcomes of the elderly in primary health institutions during study period in Qidong County, Jiangsu Province, 2014 to November 2017.

Indicators	2014	2015	2016	2017
Number of hospitalizations	19,076	24,092	23,169	25,109
Average LOS (bed-days)	9.28	8.16	8.49	7.99
Daily hospitalization costs per patient, monthly average (RMB)	434.10	440.36	456.74	518.29
Average OOP expenses, monthly average (RMB)	1829.86	1387.62	1353.75	1221.73
NRCMS reimbursement rate (%)	53.29	61.09	64.89	70.35

**Note.** LOS, length of stay. OOP, out-of-pocket. NRCMS, new rural cooperative medical system. RMB, renminbi, Chinese currency.

**Table 3 ijerph-16-04316-t003:** Results of Augment Dicky-Fuller unit root test and stationarity of outcomes.

Indicators	Intercept and Trend	ADF Statistic	*P* Value	5% Critical Value	Stationarity
Number of hospitalizations	intercept	−3.409	0.011	−2.944	stationarity
Average LOS	trend	−6.839	<0.001	−2.941	stationarity
Daily hospitalization costs per patient	trend	−6.038	<0.001	−3.516	stationarity
Average OOP expenses	trend	−3.415	0.049	−2.941	stationarity
NRCMS reimbursement rate	intercept and trend	−3.536	0.048	−3.513	stationarity

**Note.** ADF, Augment Dicky-Fuller. OOP, out-of-pocket. NRCMS, new rural cooperative medical system.

**Table 4 ijerph-16-04316-t004:** The changes in trend and level estimated by ITS regression analysis after correcting for autocorrelation.

Indicators	Slope: Jan 2014 to Dec 2014 (SE)	*p*	Change in Level: Predicted Value of Jan 2015 Based on Post-FCEMs Trend Minus Predicted Value Based on Pre-FCEMs Trend (SE)	*p*	Change in Slope from Jan 2014-Dec 2014 to Jan 2015-Nov 2017 (SE)	*p*	Parameters of Model Fit		
							DW	Root MSE	R^2^
Number of hospitalizations	−55.953 (45.602)	0.227	720.440 (329.545)	0.034	58.328 (48.691)	0.238	1.790	351.420	0.209
Average LOS	0.372 (0.082)	<0.001	−3.257 (0.625)	<0.001	−0.369 (0.083)	<0.001	2.267	1.190	0.508
Daily hospitalization costs per patient	−6.263 (2.793)	0.030	13.844 (21.478)	0.523	9.382 (2.841)	0.002	2.028	35.765	0.534
Average OOP expenses	38.035 (11.555)	0.012	−626.905 (90.261)	<0.001	−43.215 (14.950)	0.006	1.891	106.130	0.741
NRCMS reimbursement rate	0.066 (0.165)	0.692	5.301 (1.033)	<0.001	0.283 (0.185)	0.135	1.636	1.012	0.915

**Note.** FCEMs, full coverage of essential medicines. LOS, length of stay. OOP, out-of-pocket. NRCMS, new rural cooperative medical system. SE, standard error. Root MSE is the estimate of the standard deviation of residual error. R^2^ indicates the proportion of total variation ascribed to the model fit. The Durbin-Watson statistic lies between 0 and 4; generally, the closer the Durbin-Watson statistic is approaching 2, the stronger rejection of autocorrelation.

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
