# Peer review of "The Effect of the Full Coverage of Essential Medicines Policy on Utilization and Accessibility of Primary Healthcare Service for Rural Seniors: A Time Series Study in Qidong, China"

_ijerph, 2019, doi:10.3390/ijerph16224316_

Round 1

Reviewer 1 Report

This study is written well, however, I could not understand Figure 1. The changes in slope and level estimated by an ITS model.

So, this need to revise easy to understand for the readers.

I thought that there are several more limitations in the methods and discussion of the present study.

Authors also showed in the methods that "The raw NRCMS’ claims data on the inpatients aged 70 years or older in primary health 164 institutions (including all the rural community health service centers and township hospitals) was 165 retrieved from the database of the department of the new rural medical scheme in Qidong County."

In addition, authors also showed that there were several limitations in the present study like this.
"Fourth, only the elderly inpatients aged 70 years or older are selected as the target population, limiting generalizability to the entire healthcare systems."

However, I did not understand this selection of the subject's age.

Why was this study limited to those over 70 years old?
Why is age restriction?
Isn't there a restriction on age?

The author need to add the reason of these inn the method or discussions more exactly.

Author Response

Response to Reviewer 1 Comments

Point 1: This study is written well.

Response 1: We appreciate your comments, which will encourage us to do better job in the future research. Thank you very much for your useful suggestions and comments, which guided us to improve the quality of this article. We have revised the article following your opinions using the “track Changes”. We look forward to your future review.

My comments are:

Point 2:  I could not understand Figure 1. The changes in slope and level estimated by an ITS model. So, this need to revise easy to understand for the readers.

Response 2: According to your requirements and in order to make it easy to understand for the readers, we have revised Figure 1 as follows. Figure 1 is a graphic illustration of policy effects measurements in segmented regression of interrupted time series analysis. Compared to the previous Figure 1, some necessary notes, such as trend line and slope of trend line, were added on the new version of Figure 1. In addition, we have added some graphic illustrations such as “cd” and “ab” in Figure 1, which can measure the policy effects based on trend lines for data points before and after the policy. (see Figure 1).

Point 3: I thought that there are several more limitations in the methods and discussion of the present study. Authors also showed in the methods that "The raw NRCMS’ claims data on the inpatients aged 70 years or older in primary health institutions (including all the rural community health service centers and township hospitals) was retrieved from the database of the department of the new rural medical scheme in Qidong County." In addition, authors also showed that there were several limitations in the present study like this. "Fourth, only the elderly inpatients aged 70 years or older are selected as the target population, limiting generalizability to the entire healthcare systems." However, I did not understand this selection of the subject's age. Why was this study limited to those over 70 years old? Why is age restriction? Isn't there a restriction on age? The author need to add the reason of these in the method or discussions more exactly.

Response 3: Thank you for the valuable comments on the manuscript. We apologized that the targeted beneficiaries were not described explicitly in the section of Method and Discussion. Due to the local policy provisions, there is a restriction on age of beneficiaries in this study. The objective of this study was to evaluate the effects of the FCEMs policy on health service utilization and accessibility, therefore, only the targeted beneficiaries of the FCEMs policy were included in the study. According to the FCEMs policy documents that were issued by the government of Qidong County, only the rural seniors aged 70 and older were the targeted beneficiaries of the policy. As a result, the subjects of this study were limited to those patients over 70 years old.

We have added the reason of the age restriction in the section of Method as follows: According to the FCEMs policy documents that were issued by the government of Qidong County, only the rural seniors aged 70 and older were the targeted beneficiaries of the policy. As a result, the subjects of this study were limited to those elderly patients over 70 years old. (Line 129-134).

In addition, we also have rephrased the limitations as follows: Fourth, due to the local policy provisions, there is a restriction on age of beneficiaries in this study. Only the elderly inpatients aged 70 years or older are selected as the target population, limiting generalizability to the entire healthcare systems. (Line 392-394).

Reviewer 2 Report

This is an interesting paper on a health care reform providing free access to essential medicine on health services utilization, costs, and type of payment in elderly people in Qidong, China.

I suggest the following improvements be made before the paper can be considered for publication.

Extensive editing of the English is required, possibly by a native speaker. You should also try to use less technical language to explain analysis and results to readers not familiar with this type of analysis. E.g. instead of speaking of ascending and descending slopes, you could just say that OOP increased before the reform but decreased afterwards. Line 112: this is not a case study in the sense this term is usually used in medical research. It is certainly a limitation that in your calculation of average expenditure you did not include LoS; given that LoS declined after the reform you underestimate costs. Please better explain the sampling and where the data come from. E.g. which types of hospitals and how many provide data? How in terms of coverage of the county? Try to explain a little bit more about the type of data analysis you do. This is now very technical and difficult to understand for non-statisticians. I don’t understand line 182f and I have never heard of a counter variable (you may mean count) although I have been trained in biostatistics. Also, 192 about P is difficult. Moreover explain ADF and Durbin Watson to the readers in analysis and results (216ff). Table 4: How was corrected for autocorrelation? In the discussion you state that LOS for patients with less severe illness declined after reform, any evidence? You may also want to discuss that LoS declined in conjunctions with increasing number of hospitalizations. Maybe also patients could be discharged without having to pay for medicines, before they may have preferred to stay in hospital to have access to medicines. L 337ff This whole stuff with moral hazard seems mere speculation. In the conclusion you say LoS was increasing but it was decreasing after reform.

Round 2

Reviewer 1 Report

This article will be published.